

# Covariations between personality behaviors and metabolic/performance traits in an Asian agamid lizard (*Phrynocephalus vlangalii*)

Jingfeng Chen, Yin Qi, Yayong Wu, Xiaocui Wang and Yezhong Tang

Chengdu Institute of Biology, Chinese Academy of Sciences, Chengdu, China

## ABSTRACT

Ecological factors related to predation risks and foraging play major roles in determining which behavioral traits may mediate life history trade-offs and, therefore, the pace-of-life syndrome (POLS) structure among behavioral, physiological, and life-history traits. It has been proposed that activity/exploration or risk-taking behaviors are more likely to impact resource acquisition for organisms (individuals, populations, and species) foraging on clumped and ephemeral food sources than for organisms foraging on abundant and evenly distributed resources. In contrast, vigilance or freezing behavior would be expected to covary with the pace of life when organisms rely on food items requiring long bouts of handling. Nevertheless, it remains unclear how general this pattern is. We tested this hypothesis by examining the associations between exploration/risk-taking behaviors and metabolic/performance traits for the viviparous agamid lizard, *Phrynocephalus vlangalii*. This species forages on sparse and patchy food sources. The results showed positive correlations between exploration and endurance capacity, and between bite force and risk-taking willingness. Our current findings, in conjunction with our previous work showed no correlations between freezing behavior and performance in this species, support the idea that behaviors in life-history trade-offs are natural history-dependent in *P. vlangalii*, and provide evidence that behavioral types play functional roles in life history trade-offs to supporting POLS hypothesis.

## INTRODUCTION

Phenotypic integration or correlation can result from evolutionary constraints, under which multiple traits are functionally connected such that no trait may respond independently to natural selection (*Sinervo & Svensson, 2002*). Theorists have proposed that life-history characteristics and suites of physiological (metabolic, immunological, and endocrine) traits may coevolve as a function of species- or population-specific environmental conditions forming a pace-of-life syndrome (POLS) (*Réale et al., 2010*; *Ricklefs & Wikelski, 2002*). More recently, repeatable differences in behavior (i.e., "animal personality") and performance traits have been integrated in POLS research (*Careau et al.,*

Corresponding author
Yin Qi, qiyin@cib.ac.cn

*2008*; *Wolf et al., 2007*). Accordingly it has been hypothesized that high activity, superficial exploration, and risk-taking willingness are positively correlated with locomotor performance and fighting ability (*Careau & Garland, 2012*). This prediction is based on the rationale that individuals with faster pace-of-life must perform better during foraging and territorial defense to satisfy the energetic demands of their behavioral strategy (*Farwell & McLaughlin, 2009*; *Garland, 1999*). Nevertheless, the results obtained from empirical studies on the links between metabolism/performance and behaviors have been mixed (*Mathot et al., 2013*; *Royauté et al., 2015*).

It is possible that mixed results in this field occurred because the researchers did not properly choose the behavioral type: behavioral traits theoretically likely to be associated with the pace of life are determined by the type of resources an organism acquires and the foraging behavior of its predators (*Dammhahn et al., 2018*; *Montiglio et al., 2018*; *Salzman et al., 2018*). For example, activity/exploration or risk-taking behaviors are more likely to impact resource acquisition in organisms (individuals, populations, or species) foraging on clumped and ephemeral food sources than those foraging on abundant and homogeneously-distributed resources. Vigilance or freezing behavior are instead more likely to covary with the pace of life when organisms rely on food items requiring long bouts of handling (*Montiglio et al., 2018*). Thus far, however, the natural history-dependent behavior-metabolism/performance co-adaptations have rarely been investigated.

The Qinhai Toad-headed agama, *Phrynocephalus vlangalii*, is a small viviparous lizard living in northwestern China (*Zhao, 1999*). They usually forage in open spaces in arid or semiarid regions covered by sparse and patchy vegetation in the Qinghai-Tibetan Plateau (*Qi et al., 2012*). The high-elevation individuals of *P. vlangalii* grow faster than individuals of the same species living at lower elevation (*Lu et al., 2018a*), consistent with higher potential food availability and higher active body temperatures at the higher altitude (*Lu et al., 2018b*). Therefore, based on the rationale of natural history-dependent behavior-metabolism/performance coadaptation (*Montiglio et al., 2018*), we hypothesized that exploration/risk-taking behavior, but not vigilance or freezing behavior, should coevolve with pace of life traits (metabolism and performance) in *P. vlangalii*. Our previous studies have demonstrated that freezing behavior did not correlate with locomotor performance (maximal endurance and sprint speed), consistent with this hypothesis (*Peters et al., 2016*; *Qi et al., 2012*, *2014*). In the present study, we firstly examine the association between exploration propensity/risk-taking and metabolic rate/performance traits in male *P. vlangalii* and test the coadaptation among these traits as suggested in POLS, then we compared current results and previous work and see how behavioral types play functional roles in life history trade-offs to supporting POLS hypothesis.

## MATERIALS AND METHODS

### Animal sampling

All lizards were adult males (average snout-vent length (SVL) = 66.58 ± 5.68 mm, $N = 47$) from the Zoige Wetland Nature Reserve, in southwestern China (*Qi et al., 2012*). The subjects were caught by hand or in pitfall traps in early July 2015. A total of

13 individuals were captured from site A (latitude, 33°42′57.7″N; longitude, 102°29′23.4″E; altitude, 3,471 m a.s.l.), and another 34 individuals were obtained from site B (latitude, 33°44′31.5″N; longitude, 102°30′15.9″E; altitude, 3,454 m a.s.l.). At both sites, the sandy substrate is similar but the vegetation density varies across sites (it was one-fold higher at site B). The vegetation is predominantly composed of *Dracocephalum heterophyllum* and *Carex aridula*, occasionally accompanied with *Astagalus sutchenensis*, *Anaphalis lacteal*, *Vicia cracca*, *Morina kokonorica*, *Oxytropis glabra*, *Linum stelleroides*, and *Clematis tangutica*. After capture, all lizards were transferred to the Xiaman Conservation Station. Captured lizards were individually housed in card-boxes (40L × 30W × 20H cm) with sand substrate from the field. Water and mealworms were provided ad libitum. To approximate conditions in the field, we moved the boxed outside and basked the lizards when the weather allowed. All lizards were collected in 4 days. To minimizing the stress response of capturing, we housed all individuals for 5 days before we began the assessments. At the Xiaman Conservation Station in the Zoige Wetland Nature Reserve, we completed the performance measures in 2 days at first, then did the behavioral tests in another 2 days. Finally, all individuals were sent to the Chengdu Institute of Biology for 3-days metabolic tests. All animal procedures in this study were carried out in accordance with and approved by the Animal Care and Use Committee at the Chengdu Institute of Biology, Chinese Academy of Sciences (20151220) and the field experiments were approved by the Management Office of the Zoige Nature Reserve (20150701).

## Morphology and performance measurement

As we wished to evaluate how body size covaries with behavioral, metabolic and performance traits, we measured the SVL to the nearest mm and body mass, to the nearest mg, of each lizard. Each lizard was marked with acrylic paint on the dorsum to facilitate identification. Due to the close link between performance and temperature in ectothermic species (*Qi et al., 2014*), we standardized temperature during physical testing. The subjects' bite force was measured at room temperature (15 °C) by inducing each lizard to bite the free end of a steel bite-plate connected to an isometric force transducer (S1-100NHL, 0–100N; Nanjing Bio-inspired Intelligent Technology Co. Ltd., Nanjing, China). Bite forces were measured three times per individual, and the maximum value was used in further analysis (*Lailvaux, Gilbert & Edwards, 2012*). Second, the subjects were warmed with an infrared heater before measuring endurance and sprint speed in order to ensure the lizards were active and to standardize body temperature (22–35 °C). Detailed protocols for measuring sprint speed and endurance have been described previously (*Qi et al., 2014*). Briefly, lizards were stimulated to run by tapping the base of the tail with a paintbrush (*Noble, Fanson & Whiting, 2014*). All runs were recorded using a camera (HDC-HS60; Panasonic, Kadoma, Japan) and sprint speed was calculated as the fastest speed along a 1.5 m racetrack. To insure that the lizards experienced environmental conditions similar to those in the wild, the racetrack was covered with sand from the field. Endurance was measured immediately following after sprint speed tests. Lizards were placed in a circular racetrack and stimulated to run by tapping the base of the tail using a paintbrush. We considered the subject as exhausted if the lizard did not run after tapping

the tail 10 times. Total movement time before exhaustion was regarded as a proxy of endurance. Sprint speed and endurance were measured once per day over 2 consecutive days. We used the maximum value for subsequent analyses, because lizard performance can be highly variable due to differences in motivation and other factors (*Losos, Creer & Schulte, 2002*).

## Personality measurement

Two behavioral traits, exploration and risk-taking, were measured and used as proxies of the lizards' personalities. Similar traits have been measured in many lizard species and have been shown to be consistent across different contexts (*Cote & Clobert, 2007*; *Lopez et al., 2005*; *Stapley & Keogh, 2004*). All lizards received risk-taking measurement after exploration tests.

### Exploration

Lizards were individually introduced into a novel enclosure (60L × 44W × 37H cm) and were initially placed in a opaque plastic box in the center of the enclosure (Fig. 1A). Two polyvinyl chloride (PVC) refuge burrows (10L × 4R cm) were set at each end of the enclosure; these served as the goal of lizard exploration. After a 2 min acclimation period, the plastic box was lifted, exposing the lizard to the novel enclosure. Over 20 min we scored the following aspects of the subject's behavior: (i) time in locomotion and (ii) latency to enter each of the two PVC refuges in the enclosure (time to enter refuge). The time to enter refuge was taken as measurement of quickness to explore a novel environment, and time in locomotion was taken as a measurement of exploration intensity.

### Risk-taking

Risk-taking is the propensity to take risks, especially in novel situations, and is usually measured experimentally in relation to anti-predatory behavior or individual responses to novel cues. In this study, we assayed risk-taking in an anti-predatory context. The trial was carried out in the same enclosure as the exploratory assay, however the refuge arrangement was different. We randomly assigned one of the two PVC burrows as a basking site by suspending a 25W incandescent bulb (special heat bulb for reptiles) ca 15 cm above at the burrow entrance, while the other PVC burrow was made cold by surrounding it with ice (Fig. 1B). As for the test of exploration, the lizards were introduced into the enclosure in the opaque plastic box, and allowed 2 min for acclimation. Once the trial began, we removed the plastic box and allowed lizards 15 min to discover the basking burrow. We then simulated a predatory attack by chasing each lizard off the basking refuge until it entered the cold burrow. The time it took for the lizards to return to the basking sites (i.e., risk-taking intensity) was measured as a proxy of lizard risk-taking. Thus, a higher intensity index means lower risk-taking willingness.

## Metabolic capacity measurement

Minimum metabolic rate represent the maintenance costs from metabolically active organs such as gut, intestines, and liver (*Biro & Stamps, 2010*). The minimum metabolic rate at a specified temperature is usually called standard metabolic rate (SMR) for

a)

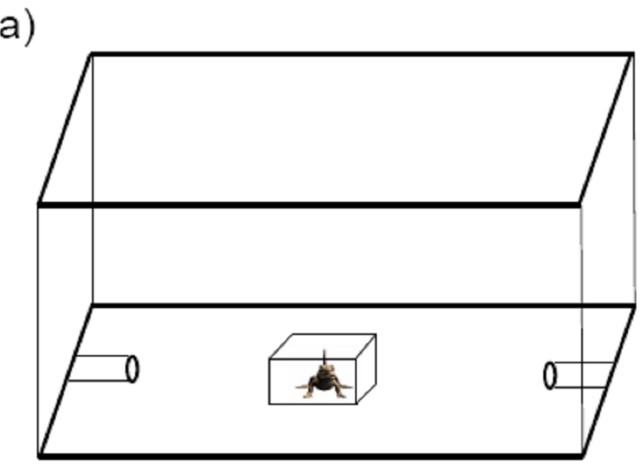

b)

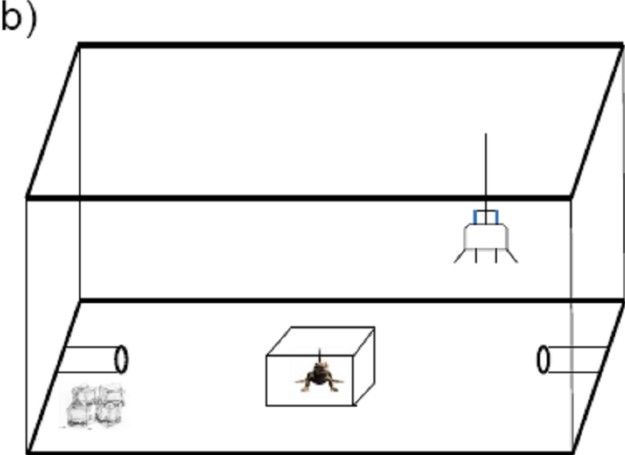

**Figure 1 Schematic diagram of the apparatus used for personality measurement in the toad headed agama _Phrynocephalus vlangalii._** (A) Exploration measurement. Lizards were initially set in an opaque box in the center. Two min later, the box was lifted and the trial was run for 20 min. Time in locomotion and time to enter either PVC refuge are counted as the intensity of exploration. (B) Risk-taking measurement. A 25W incandescent bulb was suspended close to one side of PVC burrow as basking site, while the other PVC burrow was made cold by surrounding it with ice. The lizard was acclimated for 2 min under the masking box. Then the box was removed, and lizards were allowed to approach the basking site in 15 min. Then an experimenter chased each lizard off the basking refuge until they entered the cold burrow. The time to then return to their basking sites was used to evaluate the risk-taking level.

ectotherm, and basal metabolic rate for endotherm is named when it is measured in thermal neutral zone. These maintenance costs may profoundly affect individuals' behaviors. Individuals with a higher metabolic rate could potentially afford a faster pace of life, because a higher metabolic rate allows them to mobilize the energy needed to express a high level of activity, a fast exploration, or high levels of aggressiveness. Minimum metabolic rate can be also considered as "competitor" because it monopolizes energy that would be otherwise available for other energy-demanding activities (_Biro & Stamps, 2010_; _Careau et al., 2008_). For these reasons, we used SMR as a proxy for energy metabolism to test the coadaptation hypothesis.

The oxygen volume consumed by animals provides a convenient and readily measured estimate of their metabolic rates. We measured lizard oxygen consumption using an open air flow respirometer from TSE Systems (Bad Homburg, Germany) comprised of a source of pressurized gas (14% $O_2$, 86% $N_2$, this oxygen pressure is equal to the oxygen concentration at their habitats), Low Pressure Regulator G261, eight Channel Gas Switcher G244, Sample Chambers, eight Channel Gas Flow Controller and Monitor G246, Differential Oxygen Analyzer S104 and $CO_2$ analyzer S157 with Gas Pump P651. After 2 days' fasting, the lizard was placed individually in a 190 ml darkened chamber. To standardize the measuring condition, we controlled the airflow at 110 ml/min and set the temperature at $25 \pm 0.5\,°C$. Oxygen concentration was measured for 300 s every hour and the measurement for each individual last 6 h, and the animals were weighed once at the end of the 6 h's measurement. We calculated the oxygen consumption as the difference between the oxygen concentration in the ambient air and at the chamber exit. The equation was follows:

$$VO_2 = \frac{FR * (F_iO_2 - F_eO_2) - (F_eO_2 * VCO_2)}{1 - F_eO_2}$$

Where FR is the flow rate, $F_iO_2$ and $F_eO_2$ are the fractional concentration of $O_2$ entering and exiting the respirometry chamber, respectively, and $VCO_2$ is the rate of $CO_2$ production. The minimum consumption based on six time points was used to calculate the SMR (ml $O_2$/h).

## Statistical analyses

All data were analyzed using R software version 3.2.2 (*R Core Team, 2013*) and SPSS 19.0 (IBM Corporation, New York, NY, USA). Five individuals of the 47 lizards were excluded from further analyses, because they were highly active and did not meet the SMR measurement assumption (*Kristín & Gvoždík, 2012*). A high proportion (33/42) of the remaining lizards did never entered either of the two refuges during the exploration test. For this reason, we regarded the time to enter refuge as a binomial variable, with lizards that entered the refuge assigned as "1," and lizards that did not entered assigned as "0." Similarly, a high proportion (25/42) of the lizards did not return to the basking site during the risk-taking test after the simulated predator attack. For this reason, we regarded risk-taking intensity as a binomial variable, with lizards that returned to the basking site assigned as "1," and lizards that did not return to the basking site assigned as "0."

All continuous variables were normalized (SMR and endurance were log-transformed and time in locomotion was square root-transformed) and standardized (mean = 0, SD = 1) to facilitate the following analyses.

We used the residual of the regression of performance against body temperature if we detected any significant correlation between them. We examined the correlations among behavioral traits (exploration and risk-taking), body size (body mass and SVL), metabolic rate and performances (sprint speed, endurance, and bite force) to verify the relationship between behavior, physiology, and performance predicted by the POLS.

**Table 1 Correlations between morphological, behavioral, metabolic, and performance traits in the Qinhai Toad-headed lizards.**

| | Log SMR | Bite force | Residual bite force | Sprint speed | Log endurance | Sqrt_time in locomotion | Time to enter refuge | Risk-taking intensity | Body mass |
|---|---|---|---|---|---|---|---|---|---|
| Bite force | 0.09, 0.55 | 1 | | | | | | | |
| Residual bite force | 0.14, 0.37 | 0.93, 0.00 | 1 | | | | | | |
| Sprint speed | 0.12, 0.46 | 0.14, 0.38 | 0.10, 0.55 | 1 | | | | | |
| Log endurance | 0.00, 0.99 | 0.14, 0.36 | 0.07, 0.67 | 0.03, 0.86 | 1 | | | | |
| Sqrt_time in locomotion | 0.06, 0.70 | −0.02, 0.91 | −0.03, 0.85 | 0.02, 0.91 | **0.34, 0.03** | 1 | | | |
| Time to enter refuge | −0.02, 0.89 | 0.20, 0.23 | 0.19, 0.23 | 0.01, 0.96 | −0.27, 0.09 | **−0.32, 0.03** | 1 | | |
| Risk-taking Intensity | −0.02, 0.89 | **−0.33, 0.03** | **−0.31, 0.04** | −0.06, 0.72 | −0.11, 0.48 | 0.07, 0.656 | −0.04, 0.79 | 1 | |
| Body mass | −0.11, 0.48 | −0.36, 0.02 | 0.05, 0.77 | 0.04, 0.81 | 0.28, 0.07 | −0.01, 0.98 | 0.04, 0.80 | −0.30, 0.05 | 1 |
| SVL | −0.101, 0.50 | −0.36, 0.02 | 0.00 1.00 | 0.14, 0.38 | 0.22, 0.156 | 0.03, 0.87 | 0.00, 0.99 | −0.21, 0.18 | 0.87, 0.00 |

**Note:**
In each dataset, the first number is correlation coefficients ($r$) and the second is the $P$-value. Significant values are bolded.

For the correlation analyses between time to enter refuge, risk-taking intensity, and the other variables, we used Spearman correlations since these variables are binominal. For the correlation tests between continuous variables, Pearson's product-moment correlations were applied. If performance traits were significantly affected by body size (e.g., bite force), we used the residuals of the observed variable against SVL for the correlation test. All means are reported as mean ± 1 SE.

## RESULTS

We found that our two measures of exploration test were negatively correlated ($r = −0.32$, $P = 0.03$) but we found no association between risk-taking intensity and either of our two exploration intensity indexes (time in locomotion: $r = 0.07$, $P = 0.656$; time to enter refuge: $r = −0.04$, $P = 0.79$) (Table 1).

The performance measurements were not correlated with temperature (sprint speed: $r = 0.15$, $P = 0.35$; endurance: $r = 0.11$, $P = 0.49$; bite force: $r = 0.07$, $P = 0.67$). So, we did not use the residual from a linear regression of the performance trait and temperature for the following analysis. There was no significant association between performance traits (bite force, sprint speed, endurance ability) and SMR, and among the performance traits (Table 1).

Endurance ability, but no other performance measurements was correlated with our time in locomotion ($P < 0.05$) (Table 1; Fig. 2). Moreover, only bite force/residual bite force was correlated with risk-taking intensity ($P < 0.05$). In addition, body mass was marginally correlated with risk-taking intensity ($P = 0.05$).

## DISCUSSION

In this study, we found positive correlations between exploration and endurance capacity, and between risk-taking willingness and bite force in *P. vlangalii*. Similar associations have
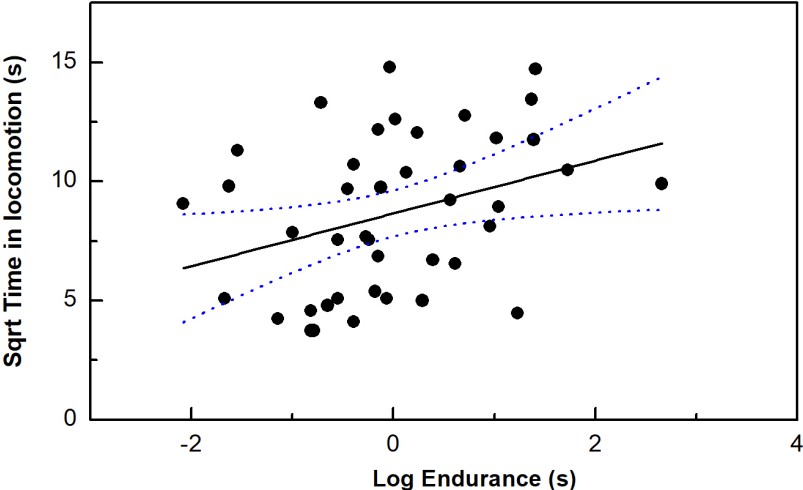

**Figure 2 The correlation between time in locomotion and endurance capacity in the toad headed agama *Phrynocephalus vlangalii*.** The dotted line represents 95% confidence intervals.

been reported in *Zootoca vivipara*, and *Tupinambis merianae* (*Herrel et al., 2009*). However, in our previous work, freezing behavior did not correlate with locomotor performance in *P. vlangalii*. Therefore, our results support our hypothesis that the coadaptation pattern is behavioral type-dependent which implies that the role of behaviors mediating life history trade-offs is associated with the natural history of the species.

Bite force variation among vertebrates has been found to correlate with measures related to fitness, including energy intake efficiency when foraging, number and quality of mates acquired, or anti-predation capacity (*Anderson, McBrayer & Herrel, 2008*). Therefore, bolder individuals with greater bite force may support a faster pace-of-life to maximize fitness payoffs. Androgens, such as testosterone, may mediate the positive correlation between risk-taking willingness and bite force, since androgens can indirectly affect performance by modifying growth rates and organizing the development of morphological traits as well as mediate plastic changes in behavior and morphology (*Coppens, De Boer & Koolhaas, 2010*; *Noble, Fanson & Whiting, 2014*).

Endurance capacity mainly results from enzymatic, physiological and morphological muscle attributes, such as body size, tight-muscle mass, and aerobic metabolic capacity (*Garland, 1984*). Because we found no correlation between body size (SVL or body mass) with endurance ability in *P. vlangalii*, the association between exploration intensity and endurance ability seems to be more likely mediated by its association with metabolic capacity. A recent study reporting that maximal oxygen consumption/metabolic scope constrain individual behavioral variation supports this idea (*Biro et al., 2018*), although the correlation SMR and endurance ability was absent in this study. It is also worth noting that body mass but not the SVL of male *P. vangalii* is marginally correlated with risk-taking intensity, suggesting individuals with better body conditions are willing to take more risk during foraging or mate searching since they are less vulnerable to predation

(*Mayer, Shine & Brown, 2016*). Similar correlations have been shown in other vertebrates (*Harris et al., 2010*; *Maillet, Halliday & Blouin-Demers, 2015*).

Our data show that exploration was positively correlated with endurance capacity in *P. vangalii*, however, SMR and exploration were not associated. This is surprising insofar as that locomotor performances represent correlates of maximal anaerobic (e.g., sprint speed) or aerobic capacity (e.g., endurance capacity) (*Jayne & Bennett, 1990*), and SMR is highly correlated with maximum oxygen consumption in general because it has been argued that selection operating on physiological factors to increase aerobic capacity and sustained activity would increase resting of metabolism as a by-product of the increase in maximal rates of metabolism (*Hayes & Garland, 1995*; *Hedrick & Hillman, 2016*). Notably, another study found the sprint speed of garden skinks *Lampropholis delicata* was not correlated with resting metabolic rate (*Merritt, Matthews & White, 2013*). A variety of evidence indicates that reptiles reply heavily upon anaerobic metabolism (*Bennett & Gleeson, 1979*). Additional studies directly examining the association between anaerobic glycolysis (e.g., lactate) and behavioral/performance traits may reveal the potential linkages. In addition to these theoretical considerations, other experimental factors may have weakened the associations obtained between metabolism and behavioral/performance traits, including low statistical power due to lack repeatable measurement in our design, or sample reduction (the five individuals potentially indicating a consistent phenotype that they were quite active and maintained high metabolic levels during the test).

## CONCLUSIONS

Overall, the co-variation between exploration and endurance capacity, and between risk-taking willingness and bite force in *P. vlangalii* support POLS, which potentially constrain independent evolution in these traits. Moreover, our study found that in *P. vlangalii* exploration/risk-taking behavior rather than freezing behavior are more important mediators of the life-history tradeoff, which is consistent with Dammhahn's idea. Further work to test the POLS may benefit from considering the role of the species' natural history in coadaptation among behavior, physiology, and life history (*Dammhahn et al., 2018*; *Montiglio et al., 2018*).

## ACKNOWLEDGEMENTS

Many thanks to Professor Steven E. Brauth in Department of Psychology, University of Maryland for language revision.

### Funding

This work was financially supported by grants from the National Natural Science Foundation of China (31370431) to Jingfeng Chen, (31572273) to Yin Qi, (31272304) to Yezhong Tang, and from the Sichuan Provincial Science and Technology Department (2018JY0617) to Jingfeng Chen. The funders had no role in study design, data collection and analysis, decision to publish, or preparation of the manuscript.

## Grant Disclosures

The following grant information was disclosed by the authors:
National Natural Science Foundation of China: 31370431, 31572273, 31272304.
Sichuan Provincial Science and Technology Department: 2018JY0617.

## Competing Interests

The authors declare that they have no competing interests.

## Author Contributions

- Jingfeng Chen conceived and designed the experiments, performed the experiments, analyzed the data, contributed reagents/materials/analysis tools, prepared figures and/or tables, authored or reviewed drafts of the paper, approved the final draft.
- Yin Qi conceived and designed the experiments, performed the experiments, analyzed the data, contributed reagents/materials/analysis tools, prepared figures and/or tables.
- Yayong Wu performed the experiments.
- Xiaocui Wang performed the experiments.
- Yezhong Tang contributed reagents/materials/analysis tools.

## Animal Ethics

The following information was supplied relating to ethical approvals (i.e., approving body and any reference numbers):

All animal procedures in this study were carried out in accordance with and approved by the Animal Care and Use Committee at the Chengdu Institute of Biology, Chinese Academy of Sciences (20151220).

## Field Study Permissions

The following information was supplied relating to field study approvals (i.e., approving body and any reference numbers):

All field experiments were approved by the Management Office of the Zoige Nature Reserve (20150701).

## Data Availability

The raw measurements are available as a Supplemental File.

## Supplemental Information

Supplemental information for this article can be found online at http://dx.doi.org/10.7717/peerj.7205#supplemental-information.

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
