# Peer review of "Covariations between personality behaviors and metabolic/performance traits in an Asian agamid lizard (Phrynocephalus vlangalii)"

_PeerJ, doi:10.7717/peerj.7205_

## Round 0.1 · original submission · Major Revisions

Three experts in your field have reviewed your article. All congratulated you on your study but all also provided detailed and thoughtful reviews in which they outline ways your article could be improved. Their reviews are thorough and so I will not reiterate their points here but I do encourage you to consider each point carefully and respond to each as you make your revisions in preparation for resubmission.

Reviewer 1 ·

Basic reporting

See my comments to the authors for details.

-English needs to be improved
-Literature should be updated
-Article structure is good
-Framework needs clarification

Experimental design

See my comments to the authors for details.

-original research within aim and scope of journal
-Framework needs clarification
-rigorous investigation
-methods are detailed

Validity of the findings

See my comments to the authors for details.

-rationale and benefit to literature needs improving
-data collected is good but statistics need revision
-single results are discussed but lacking a broader vision

Additional comments

Peer J Reviewing Manuscript 31219v1

In this manuscript, the authors aim to test the covariance between behavioral traits, metabolism and performance in a squamate lizard species, within the pace-of-life syndrome (POLS) framework.

This is a timely study on a non-model species, on a topic that has recently received ample attention in the behavioural ecology literature.

However, the study could benefit from more appropriate statistical analyses, a more complete theoretical framework, and a broader point of view in the discussion. Furthermore, the writing could be improved in some instances. I feel that all these substantial issues need to be addressed before considering the study for publication.

Here I detail my suggestions to improve the manuscript:

1. Throughout the manuscript, the authors claim to test traits associations between behaviours and metabolism hypothesized in the POLS framework. However, they miss the crucial point of the POLS: POLS theory is based on life history theory. It is the association between differences in behaviours/metabolism and differences in life history traits that should explain the maintenance of individual (behavioural) variation through life-history trade-offs. This is explained in the paper outlined the POLS theory (Réale et al. 2010, cited in the ms) and updated in the “Topical Collection Pace-of-life syndromes: a framework for the adaptive integration of behaviour, physiology and life history”, published in Animal Behavior in 2018. I suggest the authors to re-frame entirely their paper: either they actually measure life history traits, and keep the POLS framework, or they reconsider their claims and talk about testing relationships between traits involved in the POLS, but focus less on the POLS as they are actually not testing it.
2. Following this, the broad question behind the study should be clarified. One of the main questions that the field of behavioural ecology is trying to answer is: why is individual behavioural variation maintained? The association between behavior and other traits (e.g. like in the POLS) is one explanation. The introduction needs to be reworked, for now it reads as a general introduction describing what is personality, and then it jumps to describing associations with metabolic traits and performance capacity, but the general question and the implications are not clear.
3. Reading the abstract and the introduction I was under the impression that the paper would be about different populations in different ecological conditions, since the authors put emphasis on comparing POLS patterns across different ecological contextS. However, none of this is done. I suggest to keep this idea for the discussion, as it’s certainly valid and one of the main questions for the POLS field, but keep it purely speculative when they compare their species with other studies in different conditions, otherwise it’s misleading.
4. Including the framework developed in the latest papers from the Topical collection 2018 above mentioned would benefit the paper, both for the introduction and the discussion. There are several empirical papers too that could serve as example for the data analyses and how to frame the questions.
5. In regard to the behavioural measurements, from what I understand from the text, the authors did not take multiple measurements per behavior or when they did, they only kept the maximum value. “Personality” is defined as consistent among individual behavioural variation, and requires taking multiple measures per individual of the same trait in order to separate the among individual or repeatable component. Do the authors have previous data to show that their measured variables are repeatable, and thus can justify using only one measure per trait?
6. The statistical analyses are very confusing, with the high number of models described. The most straightforward analyses to do, if individuals have repeated measurements, would be to run a multivariate model in order to estimate the among-individual matrix of correlations. In any case I suggest to give some structure to the statistical analyses and the results, it’s a lot of information now with no specific order.
7. The discussion could benefit from an opening paragraph where the authors recap the aim of the study and its main findings instead of starting abruptly with discussing one by one the results.
8. The discussion is well detailed for each single finding but lacks a larger vision. I suggest adding at least one other final paragraph where the authors discuss the implications of their study, embedded in the general theoretical framework.

·

Basic reporting

This manuscripts presents an interesting, well-described study of personality, physical performance, and pace-of-life related variables in a lizard species. The experiments and analyses are explained well, and though I know that the theoretical bases for their questions are strong, the authors need make their logic and research questions more accessible to the reader.

The authors present the background to their study well in many ways. The scope is comprehensive, and the flow is, with some exceptions, logical. However, there are a number of shortcomings in the authors’ overall reporting.

In the introduction, the authors need to present the literature with more clarity, unpacking the concepts and findings that have come before them, where necessary. While the authors cite widely, the meaning of particular references is not always obvious, and this will make the logical flow of the manuscript hard to follow for many readers. In some cases, the authors ought to fill in with more content. Please see the section on general comments for line-by-line evaluations.

There is a stylistic choice that I would recommend to the authors, but I won’t insist on. Using acronyms such a SMR, SVL, TL, TER, and TRB may not be the best choice. While these acronyms fairly faithfully represent what was actually measured, they are not widely known acronyms, and due to there being four of them often used in the same context, keeping them straight is often confusing. The authors relate each of these terms to more general traits, e.g. TL is Exploration Intensity and TRB is Risk-taking. I’d recommend that the authors try using these terms instead of acronyms, and capitalization can help make these terms stand out to the reader as important variables of interest.

On the other hand, SVL, a comparatively minor variable, probably deserves to have its acronym kept, but it also is not immediately clear why SVL was measured, or what precedent there is for this measure and this correction (I don’t think there was a reference). The authors ought to think about what their most important variables are, and how to make their meaning and purpose clear to the reader, throughout the paper.

In general, the presentation of the figures was good, and the content they presented was necessary. I have comments about some of the results presented in the tables and figures.

I appreciate that the authors are non-native English speakers and that they have sought out external assistance in writing their manuscript. In general, the language is satisfactory, though I encourage the authors to continue to work with these partner(s) in future revisions of the manuscript.

The excel spreadsheet supplied appears to contain all the raw data, and could be used to replicate the authors’ results.

Experimental design

The experimental design was the strongest part of this manuscript. The authors present their methods clearly and comprehensively. Their methods ought to be sufficient for another researcher to replicate this work.

The research question is exploratory, but it is vague and not treated as such. Lines 105 - 106: “Support for the pace-of-life hypothesis is expected to be confirmed if all traits are found to be positively correlated.”

The authors did not merely correlate all their variables and draw conclusions from that. They built models, and not kitchen-sink models which are similar to correlations, but models with specific variable included, depending on the outcome variable. This implies a confirmatory framework, but the authors do not give us specific hypotheses in their intro, only that restated above. The authors need to either acknowledge that their analysis is exploratory and explain their process of exploration, or state their hypotheses more clearly so that the confirmatory framework can be used. Whichever direction the authors choose to take, they ought to further their considerations of how this work fits in/into the current literature and its gaps.

Issues arose with some of the authors’ measures – I’m referring to TER and TRB, which needed to be transformed into binary outcomes in order to be analyzed. This wasn’t what the researchers intended, and one could argue that this adds researcher degrees of freedom, but the transformation to binary is reasonable, and I don’t have a fundamental issue with it. See general comments for more.

Validity of the findings

The authors present results that are on the margin of the standard 0.05 significance level, and unfortunately for them, they are not on the lucky side of this margin. Nevertheless, the authors’ discussion is premised on positive results. This makes it difficult to evaluate the validity of their stated findings.

The authors state in their intro that will be looking for positive correlations, and in the statistical analyses section they do mention correlated variables, but they do not present the value of this correlation or others, but press on with their modeling. Some of their variables (e.g. the binary ones) do not seem particularly well suited for correlations, though non-Pearson measures like Spearman or point-biserial correlations could allow the authors to calculate and present correlations among all their variables. I would strongly urge the authors to do this. It will elucidate their results in multiple ways, and may cast light on why their p-values of interest are marginal. 0.05 is an arbitrary value, but to interpret a result that is not below the threshold, the authors need context. Correlation could give that context. So could more descriptive tables and figures.

Once the authors have reconsidered their analyses, they need to also rethink their discussion. In the meantime, one clear, positive result was the association between TL and Endurance. This is discussed relatively briefly, 4 paragraphs in. This is the authors’ strongest result; they ought to give a bit more thought to it, possibly in the context of SMR in that model.

In general, thought SMR might be associated with TRB, it is a weak association, and it seems to be not at all associated with the other two traits, TL and TER. Negative results like this are interesting! Rather than focusing on the weak positive evidence to the exclusion of the negative evidence, the authors ought to consider what it means that they found no associations where they predicted. Power? Construct validity? Ecological factors? The authors, readers, and field will benefit the most from a reasoned discussion of why expectations did not bear out.

Additional comments

The following comments are specific to particular sections of the manuscript, indicated by line number. Some are major, some are minor.
32: “Correlated”, then 33: “associated”. Why different? Are these the correct terms?
35: “Moderately” what is this word’s purpose? I don’t understand what falling “moderately on a pace-of-life syndrome continuum” means.
62 – 64: this long statement in parentheses. It needs to be unpacked, not just presented as an aside. For example, the authors ought to link lizards to ectotherms, and ideally, explain why ectotherms and endotherms metabolic rates are measured differently and set up this information for later reference. Or, move this to a more relevant location, like where the authors introduce their lizards.
66: “intuitive” – it isn’t intuitive. This needs to be setup and linked better, I didn’t find this paragraph’s logic easy to follow.
72: It was good to see the references to other models, but the authors don’t define or dig into any of these, so I while I understand the desire to acknowledge them, their mention leaves the reader wondering. If the results are discrepant, then what papers and evidence match which theories? Listing the theories and then listing some evidence unfortunately does not advance the authors’ logic.
86 – 89: In what way does the POLS hypothesis “generally propose” this? Please be more concrete. You can link this to the preceding sentence, where I also was not certain of what the “logical gap” was.
105 - 106: It is worth mentioning again, this either needs to be more explicitly exploratory or the authors need to state the hypotheses they are attempting to confirm.
121 & 123: What are the “card-boxes” or “carton boxes”? I wasn’t sure what materials these terms were referring to, so please be more specific if possible.
132: What kind of link? What is the direction of the association?
137: Why was the maximum value chosen? Is there a precedent for this? (Please cite)
149: Again, why the maximum value? The authors say that performance can be variable and do provide a reference, but I would like to see a bit more justification.
In these cases where some values are thrown out, in addition to more reasoning, I would be very interested to see correlations between measures or other descriptive data that show that the measurements are reliable and/or that the measurement chosen by the authors is empirically superior.
169: I think it should read “carried out in” ?
181: SMR – why introduce one metabolism acronym earlier, in the intro, and abandon it for this, very similar acronym later? Are they the same, or just similar? I would like to know how they are linked, if both are important enough to be given acronyms. Maybe drop the RMR acronym and just write it out?
194 – 195: “minimum” – as above, why this measurement and not the others?
198 – 200: I am intrigued by these 5 individuals. They represent 10% of the authors’ sample, and seem to have a consistent phenotype. To me, that indicates an aspect of personality that is being excluded, which is both relevant to the current analyses and interesting for future research. This should be mentioned in the discussion.
213: “links among” is too vague. The authors should say how they specified the model – what was the outcome and what were the predictors.
215: There is a test for overdispersion that I assume the authors carried out. Please give the results.
224: Where are the R-squareds? I didn’t see them mentioned anywhere else in the paper.
230: See the earlier sections for my main suggestions of how to improve the results.
239: What statistic is S? Is it named after someone? I wasn’t immediately sure what this referred to, so I think in the first instance the author should give the full name, as S is not immediately recognizable like t or df or p.
250: As mentioned, this isn’t quite what the authors found due to the issues of significance. This at least needs to be clear.
254 – 257: The bighorn sheep that Reale has studied are a great example of a high-altitude living species for which personality and POLS data have been collected. If the authors haven’t already read this work in depth, I’d suggest they take a look as there may be some relevant research for comparison and discussion in that species.
Table 1: My comments above about acronyms apply here, especially. At very least, the authors should present a caption explaining what each acronym stands for *and* what trait the measure is a proxy of. We should also know what kind of regression each model is using. Also, the authors could add the R-squareds in a bottom row as well.
Figure 1: I think the caption should explain more about what occurred, temporally, in each condition, and clarify what the ice and lamp symbols are and mean.
Figure 2: I think some text is missing from the first part of the caption: “SRM. C”
I’m not sure the plots accurately represent the models. For instance, a), the model appears to be linear, but the line is not, it has a clear curve. Why? Either the model or plot are not matched properly. Similar in c), there appears to be some curvature to the line.
There’s another, major issue in b) the association looks non-linear, which might well be why the p-value falls short. There’s a clear difference in the lower and higher SMR individuals, maybe there should be a two-piece model, or a quadratic as well as linear association, but this is an interesting twist on the data that the basic linear models completely miss. I would be very interested to hear what the authors make of this.

Finally, and distinct from the comment in the earlier section, please be consistent when using acronyms. Sometimes the authors restate acronyms (SMR, line 251), but in others, they could use an acronym they’ve established, but don’ (‘pace-of-life’, line 105).

Reviewer 3 ·

Basic reporting

The introduction works relatively well. Seems like sufficient field background/context is provided. However, the statement on line 91-93 that “few studies have directly measured all associations among personality, performance and metabolic capacity and analyze potential causal relations” is supported with 3 references (Owerkowicz & Baudinette 2008; Merritt et al. 2013; Binder et al. 2016) that did not measure “all” associations.

Figures are showing predictions instead of raw data, which is undesirable. It would be more relevant to the content of the article to show partial residuals instead of predicted values. In the end, it is not at all clear what the scatterplot figures are showing because they are not the raw data. Perhaps it’s partial residual plots, but the goodness of fit looks way too good to match the results in the table. One of the models at least on the figure appears to be the incorrect function (shape) to fit those data.

Other comments about figures: Resolution is too low. SMR acronym is not defined in the figure caption. Figure 2 states 95% standard errors, but it must be an error and should be 95% confidence intervals.

The structure of the article conforms to an acceptable format of ‘standard sections’. All appropriate raw data has been made available in accordance with our Data Sharing policy.
Lines 61-62: It makes no sense to say that animals gain and expend energy “at higher resting metabolic rates”

Experimental design

Methods are described with sufficient information to be reproducible by another investigator, except that the authors did not provide equations used for calculating SMR.

It is hard to understand how the authors analysed the data exactly. Moreover, using residuals in a second-step analysis is not recommended. Best to include body mass as a covariate in the analyses.

One of the major flaws is the design of the novel environment test. There were two refuges (lines 159-160). It is unclear if the refuges in the novel environment acted as such or as a goal of exploration? Two individuals could go strait to the refuge for different reasons: one for seeking a refuge and the other to explore it. So it is impossible to interpret this variable.

Line 139: “The subjects were warmed with an infrared heater before measuring endurance and sprint speed in order to insure the lizards were active and to standardize body temperature”, but the temperature varied between 22 and 35°C, so this is not very standardised.

Line 183-184: unclear what is meant by "within the animal’s range of activity".

Validity of the findings

Negative/inconclusive results are not treated as such. The authors “present the estimates of main effects from the full models regardless of their significance because the parameters directly relate to our hypothesis.” (lines 224-225). While this is OK, it is not OK to say that “time to enter refuge (TER) was positively associated with SMR” when the P-value of that relationship 0.86 (line 236).

The main limitation of the study is that there are no repeated measures. Repeated measures are the gold standard in the field of personality, because they allow partitioning the variance into among- and within-individual variance. It has been clearly stated that behavioural syndromes are defined at the among-individual correlations. The same applies for the POLS. Therefore, in the current study it must be assumed that the phenotypic correlations faithfully reflect the among-individual correlations.

There are many speculative comments that are not identified as such. The first is that the “The associated pattern may be related to their micro-/macro habitat” (line 253), which is based on an un-replicated two-species comparison (to be avoided at all cost). The second is the presence of a trade-off between foraging and mating, which would explain the presence of a negative correlation between risk-taking and bite force when the resources are scarce (line 279). This is a far stretch. Third, the reasoning behind the statement that motivation is a “reasonable explanation for the link between endurance and exploration” (line 290) is implicit and not fully explained. It is also not the most parsimonious explanation (perhaps high-endurance individuals are simply capable of keep moving for longer, such that their time in locomotion was higher during the novel environment test).

In the abstract, there are two sentences (lines 23-28) on how the POLS may vary depending on fitness outcomes, and therefore we need to compare patterns across populations. This is entirely misleading because the current study does not contain data on fitness or on multiple populations, such that readers will be immensely deceived when they learn they this is a study based on a single population.

Were sprint speed, endurance and bite force correlated with each other? At the moment these correlations are not presented. The best way to analyse these data would be best to use a multivariate model with all 7 traits included as response variables, and testing all possible correlations among these while conditioning on body mass. Such a model can be run using R packages like MCMCglmm and brms.

Additional comments

This manuscript is based on an interesting dataset containing behavioural, metabolic, and performance measurements. It is challenging to measure all of these traits within a given study, so the authors must be praised. However, the absence of repeated measures is a major flaw in the design. Moreover, the presence of refuges in the novel environment test is atypical and complicates the interpretation of the behaviour measured. Finally, some of the non-significant relationships are discussed as if they were significant.

---

## Round 0.2 · Minor Revisions

Thank you very much for submitting your revised article to PeerJ for review.

The three reviewers who reviewed your original submission have kindly provided feedback on your revisions. All noted that you had made considerable improvements to your article. However, all three still noted some outstanding concerns that you need to address before I can accept your article for publication.

Additionally, as the reviewers also noted that they had concerns about the clarity of the language I have reviewed your article myself and have provided tracked changes to suggest ways in which you can enhance the clarity of your writing. Additionally, I have a number of questions regarding your study that I also request you address. Please find these in the annotated pdf attached.

If you can attend to my feedback and that of the reviewers it will be my pleasure to accept your article to PeerJ.

Reviewer 1 ·

Basic reporting

English needs to be checked again by a native speaker to correct small mistakes and improve the flow of the manuscript.

Literature cited could be broader, especially general theoretical papers from the field. Some of the basic personality/POLS papers are missing in the introduction (e.g. the Réale 2010 basic POLS paper). This point was already made in the previous review.

Experimental design

The research question is sufficiently defined, but could be made more explicit in what are the specific hypothesis and predictions of the study. So far, it comes out in between the lines.

Also, the research question is not well justified by the literature. It seems to be a revised hypothesis fitted to the findings. The statements that "Activity/exploration or risk-taking behaviors are more likely to impact resource
acquisition in organism foraging on clumped and ephemeral food sources", and "Vigilance or freezing behavior is instead more likely to covary with the pace
of life when organisms rely on food items requiring long bouts of handling " are not supported. The only paper that is cited throughout (Dronne 1999) doesn't seem to have anything to do with the topic.

Once again, as stated in the previous review, it's a little confusing in which framework the study fits. It's not really testing the POLS as there are no life history trade offs measured. I do agree with the general point that POLS predictions should be more specific to the species natural history and ecological context, but little can be said in this respect if there is e.g. no comparison between populations. I think it should simply state that the goal is testing the predicted relationship between some traits involved in the POLS, and keep the rest for discussion.

The section on metabolism seems just attached on at the end of the introduction, but the rationale for including metabolism in the framework should be clarified before talking about the specific study species.

Validity of the findings

The discussion specific to the experiment and he study species is satisfactory, but to me is still missing a link to the broader implications of the findings.

The authors should also acknowledge throughout the lack of repeated measurements and that phenotypic correlations are used in place of among individual correlations.

·

Basic reporting

-English is satisfactory at this point, but could certainly be improved, and doing so would improve the manuscript overall.
- Literature coverage is good.
- Structure across sections is good, though there are a few issues within sections (particularly the intro) where some improvements should be made.
- Results and discussion follow from data and literature.

Experimental design

No additional comments on this revision, but see general comments for some related issues.

Validity of the findings

- Authors' take on their findings is more measured and comprehensive than in previous revision.
- Data collection is as good as before
- Statistics have been updated, so that they are clearer.
- Conclusions are reasonable to be drawn from these data, and speculation is minimal.

Additional comments

Line 28: "this pattern." - is there an "is" missing at the end of the sentence?

Line 67: "acquisition in organism" - I think this should read "organisms".

Lines 86 to 88: This is a long sentence and the double parenthetical expression doesn't help. It should be rephrased and/or broken up to make the ideas clearer.

Lines 73 to 96: There is a broader, related issue in the last two intro paragraphs. The final paragraph introduces SMR (twice, oddly enough), and that is about it. The previous paragraph (starting line 73) should be the final intro paragraph, because this is where the authors introduce the specifics of their study species and state their hypotheses. The authors should start by swapping these two paragraphs, and then carefully consider:
- the logical flow of the manuscript after making this change
- the order of and extent to which they introduce their key variables. Some variable details should obviously stay in the methods, but we need some information in the intro, but it needs to be logical, not just a string of descriptions of what metabolic rate or risk-taking are.

Lines 149 to 156: I appreciate the authors attempts to clarify their variables, but there are still some issues, particularly with Exploration. Exploration Intensity should not be used as the name for two different variables - adding a I or II does not in actuality make it much easier for the reader to keep track. Moreover, the authors have written "The time to enter refuges was taken as measurement of quickness to explore a novel environment, and time in locomotion was taken as a measurement of exploration intensity" which is inconsistent with the previous sentences, but makes more sense.
I suggest that the authors either call these two variables what they are "time in locomotion" and "time to enter refuge", or let "time in locomotion" be "Exploration Intensity" and "time to enter refuge" be "Quickness". But the authors should choose and remain consistent.

Lines 193 & 198: In the revised manuscript, as far as I can tell these are the first times TER and RMR are used, respectively. They are not properly defined or introduced. When the authors revise and consider variable names again, they need to keep track of all acronyms that are used, when they are first used, and where they are defined. When they are first used and where they are defined should naturally be the same.

Line 206: "produce" should be "product".

Line 211 to 222 - Results: Reporting needs to be more consistent. When the authors mention a correlation, whether it is significant or not, they should give the r or rho for the correlation coefficient and the p-value.

Lines 249 to 250: Be consistent about tense here (and all throughout- keep an eye on this). The 'was' on line 249 means the 'are' on 250 should be 'were'.

Line 253: What do the authors mean by "in general"? Please be more specific, and I would also suggest using an example or two.

Line 261: This is too short. What defines these 5 individuals? This does not need to be so much longer, but the reader will likely have forgotten about these individuals, so the authors should concisely tell us what the phenotype is.

Table 1, line 2: The authors used Pearson and Spearman correlations, so "(r)" should really be "(r or rho)" - preferably using the symbol for rho.

Reviewer 3 ·

Basic reporting

Basic reporting has been improved from the previous version

Experimental design

Methods better detailed than in previous version

Validity of the findings

In the revised manuscript, the aim is now to test the hypothesis that the temporal and spatial distribution of resources drives the relationship between behaviour and the pace of life. This is a very interesting idea that would deserve more development and empirical attention. However, the current study deals with a single population. Testing such a hypothesis would require replicated sets of populations feeding on spatio-temporally varying resources. This way, we could see if the correlations are different in the precise way predicted by the hypothesis. Since the study is based on a single population, it is not possible to test the hypothesis put forward.

Additional comments

Line 86: opposite: SMR for ectotherms and BMR for endotherms

Line 212: “We found exploration intensity I was negatively correlated with exploration intensity”

Line 239-242: The absence of correlation with SMR should be noted here.

Line 242:243: this is an interesting idea. The cited paper, however does not contain the word "metab" or "scope" or "oxyg*", so perhaps it cannot be used to support the statement.

Line 261: not clear. What are “the 5 individuals potentially indicating a consistent phenotype” exactly?

---

## Round 0.3 · accepted · Accept

Thank you for responding so thoroughly to each of the reviewers' comments as well as my own feedback. I believe your article is much improved. It is my pleasure to accept your article for publication in PeerJ.

#